# The internal realities of individuals with type 2 diabetes–Psychological disposition in self-management behaviour via grounded theory approach

Yogarabindranath Swarna Nantha[1,2]*, Azriel Abisheg Paul Chelliah[3], Shamsul Haque[2], Anuar Zaini Md Zain[1]

1 Clinical School Johor Bahru, Jeffrey Cheah School of Medicine and Health Sciences, Monash University Malaysia, Subang Jaya, Malaysia, 2 Department of Psychology, Jeffrey Cheah School of Medicine and Health Sciences, Monash University Malaysia, Subang Jaya, Malaysia, 3 Hospital Sibu, Sibu, Sarawak, Malaysia

* yoga.rabindranath@monash.edu

**Data Availability Statement:** All relevant data are within the paper and its Supporting Information files.

## Abstract

### Background

A paradigm shift in the disease management of type 2 diabetes is urgently needed to stem the escalating trends seen worldwide. A "glucocentric" approach to diabetes management is no longer considered a viable option. Qualitative strategies have the potential to unearth the internal psychological attributes seen in people living with diabetes that are crucial to the sustenance of self-management behaviour. This study aims to identify and categorize the innate psychological dispositions seen in people with type 2 diabetes in relation to self-management behaviour.

### Methods

We adopted a grounded theory approach to guide in-depth interviews of individuals with type 2 diabetes and healthcare professionals (HCP) at a regional primary care clinic in Malaysia. Twenty-four people with type 2 diabetes and 10 HCPs were recruited into the study to examine the inner narratives about disease management. Two focus group discussions (FGD) were also conducted for data triangulation.

### Results

Participants' internal dialogue about the management of their disease is characterized by 2 major processes– 1) positive disposition and 2) negative disposition. Optimism, insight, and awareness are important positive values that influence T2D self-care practices. On the other hand, constructs such as stigma, worries, reservations, and pessimism connote negative dispositions that undermine the motivation to follow through disease management in individuals with type 2 diabetes.

**Funding:** This study was funded by the MOH-NIH grant (Grant No.:91000440) received from the Ministry of Health Malaysia. YSN was the recipient of this grant. URL to grant website (http://www.nih. gov.my/web/grant-application/). The funders had no role in study design, data collection and analysis, decision to publish, or preparation of the manuscript.

**Competing interests:** The authors have declared that no competing interests exist.

## Conclusions

We identified a contrasting spectrum of both constructive and undesirable behavioural factors that influence the 'internal environment' of people with type 2 diabetes. These results coincide with the constructs presented in other well-established health belief theories that could lead to novel behavioural change interventions. Furthermore, these findings allow the implementation of psychosocial changes that are in line with cultural sensitivities and societal norms seen in a specific community.

## Introduction

Type 2 diabetes (T2D) represents greater than 90% of all diabetes subtypes, and a significant percentage of this is preventable [1]. Nevertheless, there continues to be an unabated rise in the rates of T2D worldwide over the past 2 decades [1]. By the year 2015, approximately 451 million people were afflicted by T2D globally [2, 3]. These figures dramatically surpass the modest projection made by World Health Organization (WHO) for the year 2030 by almost 2 fold [4]. The burden of T2D has incurred a tremendous strain on the national budget of many countries, prompting experts to declare T2D as a pandemic of global proportions [5].

Judging from the lacklustre trends seen in disease management spanning across the last 2 decades, a "glucocentric" approach is no longer considered a viable option in the elusive battle to turn the tides against the escalating trends of T2D seen throughout the world [5]. Studies have shown that the recommended targets for hemoglobin A1c (HbA1c) levels cannot be achieved through an arsenal of pharmacological treatments alone [6]. Instead, the effective management of T2D requires healthcare professionals (HCP) to customize an individualized strategy for each individual with T2D [2, 5, 7]. At the heart of this strategy is the sustenance of self-management practices that rely profoundly on our clear understanding of the unarticulated thought processes that define the typical behaviour of a person living with T2D.

The findings from the Disease, Attitudes, Wishes, and Needs Second study (DAWN2) suggest that the inner wellbeing of people with T2D plays an integral role in the determination of their prospective health outcome [8]. Studies also indicate optimal self-management practices are highly dependent on the effective communication between persons with T2D and their HCPs [9, 10]. In line with this postulation, there is an overwhelming consensus that HCPs should ideally be equipped with a tacit understanding of the wishes and needs of individuals with T2D [8]. Therefore, high-quality diabetes care hinges on the ability to fill the gap between research and practice by identifying the psychological needs of people with T2D.

This paper aims to expound on the self-management patterns in people with T2D closely related to an integrated behavioural model proposed in a previously published protocol [11]. Using a grounded theory approach, we aim to identify the essential drivers of self-management attitudes by exploring the internal psychological aspects that characterize the mindset of people with T2D in relation to their disease. We also intend to classify these dispositional traits of individuals with T2D into a functional behavioural model that is amenable to empirical quantitative testing [11].

## Material and methods

### Design

Detailed behavioural patterns pertinent to the objectives of this study were elucidated via a qualitative inquiry strategy. Consequently, we employed an inductive approach to 1) discover

the inner psychological disposition of individuals with T2D, 2) decipher and modify inferences based on emergent data, 3) use theory-building techniques as the foundation to draw newer insights other than those documented in literature. A maximum variation sampling strategy was used to achieve a diverse sample that corresponds to sociodemographic and disease control indicators (Tables 1 and 2) [11]. Moreover, theoretical sampling was utilized to identify information-rich cases to strengthen the validity of emergent constructs that explain self-management behaviour in persons with T2D. This study received approval from the Medical Research and Ethics Committee, Ministry of Health Malaysia (NMRR-18-151-39886) on the 30th of May 2018, and Monash University Human Research Ethics Committee on the 10th of October 2018 (Project ID: 17062).

## Participants and setting

Individuals with T2D were recruited through a purposive sampling method from the non-communicable disease department at the Seremban Primary Care centre, a regional multidisciplinary general practice clinic within the state of Negeri Sembilan, Malaysia. The process of enrollment into this project took place between May 2018 to April 2019. The inclusion criteria for people with T2D to take part in the study are as follows: 1) above the age of 18, 2) fluent in both English and Malay language, 3) evidence of a diagnosis of T2D in their case history, and 4) received followed-up at the clinic for at least 2 years [11]. The principal investigator is tasked with describing the details of the study to all individuals with T2D during their routine medical consultation at the clinic. HCPs comprised of GPs, diabetic educators (nurses), and pharmacists were also invited to join the study. Then, we obtained written consent and permission from all participants to audio record the interview sessions. Subsequently, all participants were given an appointment to attend a scheduled interview session. The demographic details of all the participants involved in this study are outlined in Tables 1 and 2.

## Data collection

Thirty-four individuals with T2D and 14 HCPs were invited to participate in the study. Four HCP declined participation, while 10 T2D people did not respond to our invitation. Ultimately, 24 people with T2D (in-depth interviews) and 10 HCPs (key-informant interviews) joined this study. Each interview lasted between 75 to 90 minutes. Once the formal coding process was complete for all the transcripts derived from the in-depth interviews, 2 FGDs (8 people with T2D in each session, each session lasting approximately 90 minutes) were conducted as part of the data triangulation process.

   All participants were interviewed in a room within the confines of the clinic that was specifically allotted for the purpose of this study. In-depth interviews were guided by research questions (compiled in the form of a topic guide; S1 File) designed to assess the inherent psychological attributes closely associated with self-management behaviour in persons with T2D [11]. Steps were also taken to ensure these interviews remained unstructured enough to discover novel concepts and ideas. Moreover, the topic guide was updated continuously in tandem with the acquisition of newer themes from these interviews. A detailed description of this process is summarized in a flowchart in S1 Fig.

   After the 10th round of interviews, the interviews did not uncover newer themes and theoretical saturation was considered to have been achieved. The validity of the emergent narratives from these themes was explored, reinforced, and finalized via additional interviews with 14 individuals with T2D. The interviews were audiotaped and transcribed verbatim. Field notes were made in the form of memoing during and after the interviews. One researcher (YSN) conducted all interviews.

**Table 1. Demographic details of T2D patients of the study.**

| Subject Characteristics | N |
|---|---|
| | |
| **Gender** | |
| Male | 15 |
| Female | 9 |
| **Age In Years** | |
| < 30 | <5 |
| 31–50 | 6 |
| 51–60 | 7 |
| 61–70 | 7 |
| > 70 | <5 |
| **Ethnicity** | |
| Malay | 12 |
| Chinese | <5 |
| Indian | 10 |
| Others | <5 |
| **Marital Status** | |
| Unmarried | <5 |
| Married | 19 |
| Divorced | <5 |
| Widow/Widower | <5 |
| **Work Status** | |
| Not working | 12 |
| Working Full-time | 8 |
| Working Part-time | <5 |
| Self-employed | <5 |
| **Level Of Education** | |
| Certificate level | 19 |
| Diploma | <5 |
| Bachelor's degree | <5 |
| **Diabetes duration (years)** | |
| < 5 | 5 |
| 6–10 | <5 |
| 11–15 | 5 |
| 16–20 | <5 |
| > 21 | <5 |
| **Number of medications** | |
| 3–5 | 12 |
| > 6 | 12 |
| **Type of medication** | |
| Oral only | 5 |
| Insulin only | <5 |
| Combination | 15 |
| **Glycaemic status** | |
| 6.5–8.0% | <5 |
| 8.1–9.0% | 9 |
| 9.1–10% | 5 |
| > 10% | 8 |

(*Continued*)

**Table 1.** (Continued)

| Subject Characteristics | N |
|---|---|
| **Complications** | |
| No | 17 |
| Yes | 7 |
| **Number of comorbidities** | |
| 1 | 13 |
| 2 | $<5$ |
| $>3$ | 8 |

## Data management and analysis

Proofread transcripts were then entered into a qualitative software (Atlas.TI qualitative analysis program, Version 7, Cincorn Systems Inc, 2008 [to code of transcripts and generate themes]; NVivo Plus, Version 12, QSR International, 2019 [to determine inter-coder and content consistency]) and coded manually to identify specific concepts. The data were coded in accordance with the classical and constructivist Grounded Theory methods [12, 13].

All transcripts were coded independently (S1 Table). A list of emerging themes and categories was then generated by the primary researcher (YSN). Subsequently, two independent coders scrutinized and coded the transcripts for categories, themes, and sub-themes. An excellent inter-coder consistency was found (Cohen's κ = 0.84). Disagreements were resolved via discussions. Similarly, the themes derived from this study were assessed by a panel of experts and showed excellent content consistency (Cohen's κ = 0.88).

## Trustworthiness

This study's trustworthiness was enhanced by using multiple data collection techniques (S1 Fig; S1 Table) such as in-depth interviews and FGDs (methodological triangulation) [14]. Two

**Table 2.** Demographic details of healthcare professionals involved in the study.

| Subject Characteristics | N |
|---|---|
| **Profession** | |
| Diabetic educator (nurse) | 5 |
| GPs | $<5$ |
| Pharmacists | $<5$ |
| **Mean Age In Years** | 34 |
| **Mean Years In Service** | 10 |
| **Ethnicity** | |
| Malay | 7 |
| Chinese | $<5$ |
| Indian | $<5$ |
| **Marital Status** | |
| Unmarried | $<5$ |
| Married | 7 |
| Divorced | $<5$ |
| **Level Of Education** | |
| Diploma | $<5$ |
| Bachelor's degree | 5 |
| Master's degree | $<5$ |

researchers were involved in data analysis and interpretation (investigator triangulation). Different sources (people with T2D and HCPs) of the same information were used to validate data (data triangulation and negative case analysis). The authors repeatedly analyzed, theorized, and revised concepts at various stages of the study (persistent observation).

## Results

In our study, the internal psychological reality of individuals with T2D can be summarized into two main constructs– 1) positive disposition and 2) negative disposition. There appears to be a dynamic interaction between these constructs and numerous other concepts shown in Fig 1 –the factors associated with the external reality and the mediators of optimal behavior create the tapestry for the workings of the internal realities seen in our participants. The balance between an array of personal dispositional qualities appears to determine compliance with self-management behaviour (Fig 1, Table 3, S1 Table).

### Positive disposition

**Optimism and positivity.**   Our study demonstrates that participants with T2D who appreciate the value of being alive also "wish to live longer" and frequently express gratefulness for not having developed any complications related to their disease. Similarly, information from the FGDs conducted in this study indicates that most T2D participants believe that a right "mental attitude" spurs better self-management practices. For example, several participants with T2D feel that they should first start by "being honest with themselves" if they sincerely wish to embody the habits that come hand-in-hand with optimal self-care practices.

> "If there is a problem with their attitude [towards the disease] then everything will take a turn for the worse. So, it depends on their attitude and how they see the sickness in themselves."
>
> (64, Female)
>
> "I want to be honest to myself for the sake of my health. Only then I can certain I will not fail or miss taking my medications." (60, Male)

Many individuals with T2D interviewed in our study adopt a strongly positive attitude towards their disease to the degree that they "feel healthy" despite having T2D. Most of them express indifference about being diagnosed with T2D and remain convinced that they can effectively manage their illness. They are confident about being well-equipped with the information required for appropriate disease management, especially in relation to medication compliance.

Many T2D participants display a favourable attitude towards modern medicine and express little hesitation about consuming medications. They also claim it takes minimal effort to comply with the recommended medication timings suggested by their general practitioner (GP) and pharmacist. Generally, insulin users in our study have positive views about using injectables and dismiss injecting insulin as being painful. These beliefs are primarily driven by the optimism that disease-related complications can be fully averted by being compliant with medications.

> "I believe 'Western'[allopathic] medications work very fast in bringing down sugar levels. For me, 'Western' medications are important because you can see from research results that science has progressed really far." (52, Male)

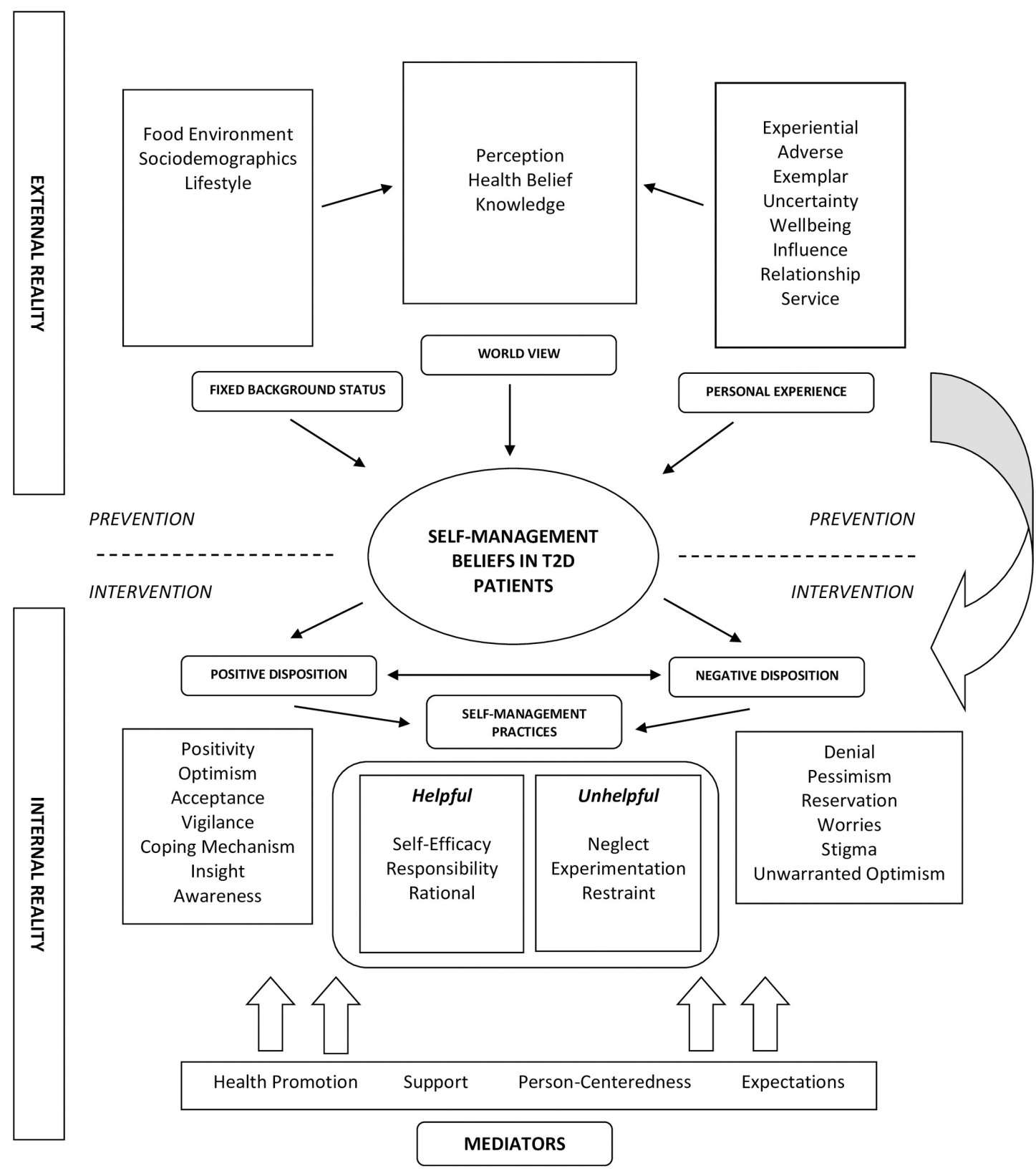

**Fig 1. Conceptual framework describing the external reality, internal reality and mediators related to self-management of T2D.**

**Table 3. Categories, definitions, frequency, and codes describing the personal disposition of individuals with T2D.**

| Category | Sub-category | Frequency | Codes |
|---|---|---|---|
| **Positive disposition** | Insight | 285 | Clarity about the role of insulin |
| | | | Taking charge of health |
| | | | Understanding the need for medications |
| Positive innate | | | Knowing the nature of disease |
| qualities of the | Optimism | 260 | Attitude towards medication |
| mind and | | | Attitude towards disease |
| character | | | Innate attitude |
| | Awareness | 200 | Specific knowledge about disease |
| | | | Diet and exercise |
| | Acceptance | 88 | Accepting illness |
| | | | Accepting the need to take medications |
| | Vigilance | 43 | Being cautious |
| | Positivity | 42 | Having the right mindset |
| | | | Appreciating the value of life |
| | Coping | 50 | Being true to self |
| | mechanism | | Positive thoughts |
| | | | Being spiritual |
| **Negative disposition** | Reservation | 181 | Negativity about having the disease |
| | | | Reluctance to use medication |
| | | | Issues with medication compliance |
| Negative | | | Relationship with doctor |
| innate qualities | Worries | 174 | Effects of complications on life |
| of the mind and | | | Fearing complications |
| character | | | Concerns about medications |
| | | | Concerns about illness |
| | | | Concerns over the behaviour of doctors |
| | Stigma | 141 | Discrimination |
| | | | Avoidance |
| | Pessimism | 130 | Unhelpful thoughts about taking |
| | | | medications |
| | Unwarranted | 55 | Curability of disease |
| | optimism | | Violations in dietary restrictions |
| | | | Medication avoidance |
| | Denial | 36 | Unwilling to accept reality |

"Definitely no [developing complications] doctor. If I take my medications accordingly and follow the advice of the doctors, I foresee no problems." (42, Male)

**Acceptance.** HCPs such as GPs, pharmacists, and nurses confess that it takes time for participants with T2D to grasp that they need to live with T2D for the rest of their lives. Grappling with the immediacy of reality, they eventually accept the idea that T2D has occurred in them with a sense of partial resignation. They profess they have "come to terms" with the realities of their disease. They quickly move on to develop the mental fortitude required to cope with the sudden changes to their ordinary way of living.

"At the end of the day, you have to overcome it yourself. You need to reach some kind of understanding. First, you have to come to terms with yourself. Look, you have diabetes and

you've got this sickness. So the only way to make you better is to stick with your medications." (44, Male)

With the passage of time, we found that the acceptance of their disease cultivates better medication compliance in T2D participants. They no longer struggle with the initial ambivalence related to frequent medication non-adherence. They become unperturbed about taking medication in the presence of others (especially injecting insulin) and easily conquer the initial fear associated with insulin utilization.

"I remember being late one day, I had to stop by at the petrol station. I got out [of the car] and started to use my insulin. A few of my friends were coincidentally there too. They asked me 'what's this?' and I answered [nonchalantly] 'well it's insulin'. So no worries there." (59, Male)

**Insight.** Participants with T2D perceive diabetes as a chronic disease that can only be controlled and will "always remain in our blood". They consider the presence of physical symptoms an unreliable indicator of poor T2D control partly because these warning signs remain imperceptible while having high blood glucose levels. Therefore, they often yearn to control their disease better, knowing full well complications can occur due to poorly controlled blood glucose levels.

T2D participants place a high degree of faith in the efficacy of medications prescribed to them; some even confess that "it will really bring down your sugar levels". They are acutely aware of the dangers of non-compliance and realize that it is imperative to consume medications as advised. Additionally, they believe that taking medication in accordance with the recommended timing schedule does play a crucial role in stabilizing blood glucose levels.

Individuals with T2D in our study believe insulin is a far potent alternative when compared to oral medication and clearly indicate a greater preference for insulin over oral medications. They also often become more receptive to the idea of insulin commencement once their GPs successfully convince them to do so.

"Why should I take insulin? And then one day came along this doctor, advising me that it is good for me. Now I feel healthier [after starting insulin]" (70, Male)

As far as lifestyle modification is concerned, most T2D participants find the adherence to dietary restrictions burdensome only at the preliminary phase of their disease. Moreover, they believe these dietary measures are critical to the long-term control of T2D.

"Initially, when I was told to control my diet, I thought it was missing it all [food]. But that was only in the beginning. Soon, I became accustomed to it." (60, Male)

"You see, everything you consume will affect your sugar levels. You have to understand that. So there is a limit to everything. If you don't control it [food], your sugar levels will increase". (52, Male)

**Awareness.** From our interviews, having adequate knowledge about the disease fosters better awareness in T2D participants, which in turn closely mediates the "willpower to change themselves". Therefore, well-informed T2D participants acknowledge that optimal disease control requires both medication compliance and the sustenance of therapeutic lifestyle changes in equal measure. They believe that it is necessary to consume lifelong medications despite having learned medications can, at times, cause certain undesirable side effects.

Participants with T2D recognize the importance of following appropriate dietary measures to manage their disease and often attribute uncontrolled blood glucose levels to poor dietary habits. They are cognizant of the types of food that are suitable for consumption in individuals living with T2D. For example, the intake of refined sugar is considered "dangerous". Therefore, most interviewees with T2D squarely blame the overconsumption of sugar (e.g. drinking sugary beverages) as the principal cause of T2D. They believe that an indulgent lifestyle (lacking both dietary restraint and adequate exercise) is most likely the root cause for having acquired T2D in the first place.

> "For me, diabetes is self-inflicted. Because I feel that in the past, I did certain things that might have caused me to have this disease. My parents never had diabetes. I am a diabetic because I did not have restraint over the food that I consumed." (61, Male)

Additionally, participants with T2D are highly conscious of the need for adequate physical exercise and how it could potentially help achieve better control of their disease. Many T2D interviewees believe that exercising can work as an adjunct to reducing blood glucose levels in their bodies.

> "It doesn't matter what my sugar levels might be, I will do exercise and look after my diet so that I will not experience any complications" (80, Male)

> "I love to exercise; it will bring down sugar levels in our body" (65, Female)

**Vigilance.**　Participants with T2D acknowledge that diabetes is a disease that should not be taken lightly. During FGDs, these individuals state that they remain in a state of constant vigilance—they are "more careful" and watchful for any T2D related complications. They also tend to be mindful of recommended lifestyle changes such as exercise and dietary recommendations. They frequently investigate fluctuations in their blood glucose levels using glucometers at home.

> "From what I know, diabetes is a serious sickness for me. You need to take good care of yourself to overcome this sickness." (59, Male)

**Coping mechanism.**　Most participants with T2D believe that God determines their destiny and, by default, the progress of their disease. They rely on God to provide deliverance from any anxiety that stems from the inner insecurities about the nature of their disease. Some T2D people go so far as to adopt a 'man proposes, God disposes' ideology.

> "I leave that [the way the disease progresses] to God. You do your part and ultimately God decides." (70, Male)

More secular-minded T2D participants resort to thought avoidance techniques to negate negative narratives about the prospects of their disease. These T2D participants also view life as being impermanent. As a consequence, they embrace uncertainty and live life as determined by fate.

> "Talking about kidney-related complications and all that, you know, there are many things in this world that are basically unknown doctor. Nothing is certain in life." (42, Female)

## Negative disposition

**Denial.** Nearly half of the participants with T2D in this study expressed initial difficulty accepting the diagnosis of T2D. This state of denial was reinforced by assuming that being previously "active" (due to the nature of work or engaging in sports activities) should have prevented T2D.

> "I couldn't accept reality [diagnosed with T2D]. I felt I could just go on with my life as usual. I made sure I did more exercise. I still indulged in food." (52, Male)

> "I should not have been diagnosed with diabetes. I was an athlete, so I believe I should not have this disease. I did all sorts of exercise activities. Why does this have to happen to me?" (60, Male)

**Pessimism.** Many participants with T2D feel depressed about having contracted diabetes. The abruptness in the demands for sudden lifestyle modifications imposes "too much restriction" to the conventional way of going about with their lives. This situation affects their emotions in such a way that some of these individuals think that it's not worth living any longer. This notion is especially true if they have to endure any complications as a result of their disease. After a protracted period of time, some of them descend into helplessness characterized by low morale or a "giving up" mentality. On the other hand, a small group of T2D participants remain relatively untroubled as they have surrendered to the notion that everyone will develop T2D at some point in their lives.

> "Why try so hard and control it [T2D]? Just enjoy life as it is. One way or another, you will definitely end up getting diabetes. You just need to take your medications. Life goes on." (65, Male)

The frustration of taking medications all their lives appears to dominate the thoughts of most T2D participants. They describe getting fed up with taking medication all the time and wanting to "give it a break" by going on occasional 'drug holidays'. Most importantly, they perceive taking more medications is synonymous with the deterioration in their general health status.

> "When they told me that I really need to take medication for the rest of my life, my heart literally skipped a beat. I was totally upset. I really dislike taking medications." (53, Female)

> "Sometimes you feel like 'it's ok, you don't need to take it today, just give your body a rest'. It's that kind of feeling." (28, Female)

**Reservations.** There appears to be an initial reluctance on the part of interviewees with T2D to start taking T2D medications early. GPs reveal that they receive frequent requests from T2D people to delay the commencement of T2D medications. This avoidant behaviour becomes even more apparent when GPs suggest the introduction of insulin into their medication regime. They vehemently dismiss the need for insulin by reiterating they prefer oral medications instead. Much later, when the stark realization sets in, participants with T2D slowly realize they now "have no choice" but to take their medications as advocated by HCPs.

> "But I have this sickness, I don't really have a choice. We have got to take our medications. What else can I do? We have no choice but to take our medications" (65, Female)

Compliance with medications does not appear to be an obstacle in the early stages of the disease. However, participants with T2D gradually develop conflicting thoughts about adhering to medications after a few months of living with diabetes. One prime reason for this is their constant preoccupation with the side effects of medications. Consequently, many of these individuals tend to skip their medications when side effects arise without soliciting the advice of their GPs. These adverse reactions cause an aversion to medications to which T2D participants respond by first doubting the efficacy of T2D medications and then skipping their medications altogether.

> "I have already taken my medications you know. But it still doesn't go down [blood glucose levels]. That's when I think to myself, how could this happen? That's why I skip my medications sometimes." (42, Male)

Our interviews brought to light several problem areas that appear to undermine optimal doctor-patient relationships during routine clinical consultations. Among others, participants with T2D often hesitate to consult GPs about any concerns about their disease out of the fear of being reprimanded. They are also reluctant to accept their GP's decision to modify the dosage of their current medications. Participants with T2D interpret this as a deliberate attempt at prematurely increasing the dosage of their medications.

> "They [individuals with T2D] are afraid to ask anything. The doctor might turn around and say 'who is the doctor now?'. I have experienced that before. So they [individuals with T2D] don't want to get hurt. And that's the reason why don't want to ask [questions]." (64, Female)

> "What they [GP] recommend you take once a day has now become twice a day. From twice a day, it becomes thrice a day. So, that's the problem. That's why I dislike taking medicine. There's no end to it." (65, Male)

**Worries.** Participants with T2D worry about developing T2D related complications (renal impairment, heart disease, and amputations) in the near future. Above all, they dread acquiring any form of kidney impairment as a result of their condition. They do not wish to lead a life fraught with complications as it considerably reduces their quality of life. The slightest deviation in their wellbeing (e.g., the presence of uncontrolled blood glucose levels) heightens the state of fear that they might succumb to complications. Hence, this behavior engenders a belief system where participants with T2D perceive that there is a significant risk of developing complications if they do not take medications as instructed by HCPs.

> "I have to take my medications, I am worried that something would happen to me. I could get a stroke or they might amputate my leg. That's what I am worried about" (75, Female)

Participants with T2D worry about immediate and long-term harm medications can apparently inflict on their body. A small proportion of these individuals feel uneasy about the idea of using insulin needles out of the fear of pain. Interestingly, a majority of T2D participants are apprehensive about "taking too much medication". They believe consuming medications for an indefinite length of time could lead to permanent physical impairment.

> "I have to take 7 pills in the morning, 7 in the afternoon and another 7 at night. That's a total of 21 pills in a day. These pills can somehow 'accumulate' in your vital organs. It will eventually destroy your kidney and pancreas" (60, Male)

**Stigma.** T2D participants often classify themselves as a "sick person" and feel embarrassed for having contracted the disease. A few of them think they will be "discriminated" if others discover that they have diabetes. Thus, many T2D people choose to keep their diagnosis confidential.

"You should see at their reaction and the look on their faces. They think I am going to die tomorrow. And that affects me a lot. I want to disclose my diagnosis sometimes. But I have already seen the double standards in the way they treat me previously." (42, Female)

These negative perceptions strongly affect the emotions and behaviour of these individuals at many different levels. For example, they avoid being seen while taking medications. Similarly, insulin users in our study feel depressed and worry about what others might think now that they have commenced using injectables. Interviews with pharmacists and diabetes educators reveal that these emotions trigger a certain uneasiness injecting insulin in the presence of others. Therefore, many participants who are insulin users frequently excuse themselves (preferring privacy) when administering insulin.

"I prefer to do it in private when I am alone in my room. Usually, I will wake up in the morning, shower and then I will use the insulin pen to inject myself."(61, Male)

**Unwarranted optimism.** In the earlier stages of their disease, T2D participants hold a fervent belief that the disease can be cured and wonder why advancements in science have not brought about any permanent solution. In terms of dietary adherence, these individuals legitimize violations in food restrictions so long as they keep taking their medications in an orderly manner. They also downplay the seriousness of elevated blood glucose levels. This behaviour is further reinforced by the absence of any alarm symptoms after intentionally skipping their medications for several days.

"Because when I take my medications and inject insulin, I feel I am alright, therefore I can eat anything I want." (48, Female)

"Doctors have warned me about the effects of high sugar levels in my body. But I didn't take it too seriously initially. I only started worrying when I witnessed complications in other patients much later." (28, Female)

"Sometimes, I don't take my medication as suggested by the doctor. Because you see, when you have diabetes, you don't feel anything even when if your sugar levels are elevated. You don't feel a thing." (60, Male)

## Discussion

### Summary of main findings

The findings from our study demonstrate the contrasting behavioural aspects that govern the internal psychological environment of people with T2D. The conceptual framework in Fig 1 implies that these attributes are insulated and function in a dichotomous manner. However, in reality, these behavioural attributes are mutually connected and certainly carry more merit than just the sum of its parts. As evidenced by the results of this study, these behavioural properties have a wide-ranging influence over the self-management practices of individuals with T2D and present a unique set of opportunities for focused intervention.

Two major themes were identified in relation to the cognitive reaction of participants with T2D to their disease, namely, 1) positive disposition and 2) negative disposition. Sub-categories within each theme can be further classified as constructive (positivity, optimism, vigilance, acceptance, coping mechanism, insight, awareness) and undesirable (denial, pessimism, reservations, stigma, worries, unwarranted optimism) behavioural factors. This distinctive classification makes it possible to realistically identify and cultivate more positive traits amongst people with T2D. In the long run, this step can negate the unproductive mindset seen in these individuals by 1) enabling greater personalization of pre-existing treatment strategies and 2) facilitating more sustainable self-management practices that are connected to their belief system.

## Comparison with literature

The theoretical components obtained from this study enhance and build upon other well established behavioural models commonly utilized to explain the widely held beliefs and perceptions patients have about their illness [15]. For the most part, this belief system mirrors the 5 dimensions seen in the concept of illness cognitions [15–17]. For instance, in our study, individuals with T2D shared detailed information about their relationship with their disease by recognizing the concepts of identity, consequences, controllability, and curability of T2D [15,17]. These perceptions evoke an emotional response (positive or negative disposition) in T2D people in our study. Most importantly, these reactions serve as a precursor in developing an appropriate coping strategy in relation to their disease. This chain of events is in line with Leventhal's self-regulatory model [15]. Nevertheless, what seems lacking is an appraisal phase where individuals with T2D contemplate either persisting or choosing an alternative strategy to manage their disease.

Several categories found in this study share close parallels with the cognitive adaptation theory [15]. For example, participants with T2D have a clear understanding of the nature of their illness (awareness and reservations). They often ponder the impact T2D will have on their lives (acceptance, stigma, and worries). They also display elements of self-enhancement–participants with T2D embrace positive thoughts (optimism and positivity) that help boost self-esteem. Additionally, the illusory aspects of their thought processes were exemplified by an unwarranted optimism in how they cope with their disease. The equilibrium between the various shades of these personal attributes influences the state of self-mastery (central to the process of cognitive adaptation) in these individuals [15].

## Strengths and limitations

This study depicts the summation of the various behavioural traits of the archetypal T2D people into a unified operational model. Although many separate attempts were made to describe patient self-management behaviour in Malaysia, we uniquely examined and condensed the overall behavioural attributes of individuals with T2D within a single practice-based qualitative study [18–20]. Moreover, our findings widely correspond to the results seen in other T2D people assessed worldwide, which further substantiates the opinion that most T2D people share a common psychological profile in terms of self-management behaviour [21–23].

Contrary to previous fragmentary descriptions of self-management practices in the form of systematic reviews or qualitative metanalysis [21–23], our study consolidates the connections and interactions between these behavioural attributes of a typical T2D person found in a conventional clinical setting. While many of the themes presented in this study have been described elsewhere in literature [21–23], our study collates these findings into a functional framework that defines the attributes of T2D people with regards to self-management practices

(Fig 1). Moreover, we have classified the psychosocial challenges of individuals with T2D into operational codes by adhering to a stringent set of theory-building research techniques (S1 File, S1 Table). This initiative is in tandem with the recommendations from the DAWN2 study, which calls for the formulation of a distinct psychosocial approach that is specific to cultural sensitivities and societal norms [8].

Individuals with T2D in this study were recruited from only one regional primary care institution. However, the richness and quality of the data were established through methodological triangulation, where selected HCPs from adjacent clinics were interviewed to verify the views put forward by individuals with T2D. Additionally, we emphasized inclusivity by obtaining people with T2D 1) from a community-based setting and 2) with a mixed sociodemographic background.

## Implications for future research and clinical practice

Our research initiative supports the need to transform naturalistic narratives derived from a qualitative inquiry into measurable scales that accurately reflect the behaviour intrinsic to the population that is being studied [24]. This carefully planned real-world evaluation, coupled with the subsequent process of interpretative deduction, has uncovered practical targets for behavioural change that can be easily integrated into routine clinical practice. This step (as opposed to standard care alone) could enable a more person-centered approach, leading to smoother behavioural transformation in people with T2D.

This study's overarching objective is to create a realistic benchmark for self-management behaviour in T2D people from data that is acquired from a naturalistic environment [25]. Many pre-existing scales in literature have not been rigorously evaluated in terms of validity and reliability [25]. Furthermore, these inventories lack robust psychometric properties, rendering it challenging to detect any changes in self-care practices as a result of behavioural change intervention [25]. To that end, the creation of this operational model for self-management behaviour can ultimately generate a largely cohesive psychometric inventory explaining the many different shades of emotions that characterize the psyche of individuals with T2D.

## Conclusions

An approach that prioritizes the needs of the patient (person-centered) rather than an exclusive focus on their "disease" alone could break the deadlock impeding optimal disease control in T2D people [5]. To that end, there is a window of opportunity for HCPs to first analyze the personality characteristics of T2D people as depicted in this study and subsequently formulate useful evidence-based psychological interventions [24, 26]. GP consultations guided by an awareness of these inherent strengths and weaknesses can help empower individuals with T2D to actively take charge of their disease [5, 27]. Furthermore, the behavioural constructs discovered in this study can be used to create a psychometric inventory to gauge the psychological polarity of T2D people in terms of their commitment towards self-management behaviour. This step would enable HCPs to implement and appraise the outcomes of a streamlined evidence-based intervention in many different clinical settings.

## Supporting information

**S1 File. Topic guides for in-depth interviews and focused group discussions.**
(DOCX)

**S1 Fig. Flow chart of the study design and development of conceptual framework.**
(DOCX)

**S1 Table. Themes, sub-themes, codes, quotations within the conceptual model (via grounded theory Approach).**
(DOCX)

**S2 Table. Consolidated criteria for reporting qualitative studies (COREQ): 32-item check-list.**
(DOCX)

# Acknowledgments

We are eternally grateful to all those extraordinary individuals at Monash University who continue to remind us about humility and compassion behind their random acts of kindness. We would also like to dedicate this article to Musonius Rufus, Seneca, Marcus Aurelius and Publius Syrus, whose works gave us the wisdom to condense the complexity of human emotions into legible words accessible to all.

# Author Contributions

**Conceptualization:** Yogarabindranath Swarna Nantha.

**Data curation:** Yogarabindranath Swarna Nantha, Azriel Abisheg Paul Chelliah.

**Formal analysis:** Yogarabindranath Swarna Nantha, Azriel Abisheg Paul Chelliah.

**Funding acquisition:** Yogarabindranath Swarna Nantha.

**Investigation:** Yogarabindranath Swarna Nantha, Azriel Abisheg Paul Chelliah.

**Methodology:** Yogarabindranath Swarna Nantha.

**Project administration:** Yogarabindranath Swarna Nantha.

**Resources:** Yogarabindranath Swarna Nantha.

**Software:** Yogarabindranath Swarna Nantha, Azriel Abisheg Paul Chelliah.

**Supervision:** Shamsul Haque, Anuar Zaini Md Zain.

**Validation:** Yogarabindranath Swarna Nantha, Azriel Abisheg Paul Chelliah.

**Visualization:** Yogarabindranath Swarna Nantha, Azriel Abisheg Paul Chelliah.

**Writing – original draft:** Yogarabindranath Swarna Nantha.

**Writing – review & editing:** Yogarabindranath Swarna Nantha, Shamsul Haque.

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
