## [Decision Letter · Decision Letter 0]

5 Feb 2021

PONE-D-19-33966

The internal realities of individuals with type 2 diabetes  – understanding personal disposition towards disease management via Grounded Theory approach

PLOS ONE

Dear Dr. Swarna Nantha,

Thank you for submitting your manuscript to PLOS ONE. After careful consideration, we feel that it has merit but does not fully meet PLOS ONE’s publication criteria as it currently stands. Therefore, we invite you to submit a revised version of the manuscript that addresses the points raised during the review process.

We look forward to receiving your revised manuscript.

Kind regards,

Nelly Oelke

Academic Editor

PLOS ONE

Journal Requirements:

2. Please address the following:

- Please include additional information regarding the survey or interview guide used in the study and ensure that you have provided sufficient details that others could replicate the analyses. For instance, if you developed a questionnaire as part of this study and it is not under a copyright more restrictive than CC-BY, please include a copy, in both the original language and English, as Supporting Information. In addition, please include further information concerning the development and/or pre-testing of this guide.

- Please provide the dates of patient recruitment to this study.

4. Please remove your figures from within your manuscript file, leaving only the individual TIFF/EPS image files, uploaded separately.  These will be automatically included in the reviewers’ PDF.

Additional Editor Comments:

Thank you for your resubmission of your manuscript. I have reviewed the reviewer comments and provide some additional comments for your manuscript. These are listed below. Please ensure that you provide a track-changed document in re-submitting your manuscript.

Specific comments:

Page 2, Line 6: In the abstract, “dimension” should be pluralized to “dimensions”

Page 2, Line 7: In the abstract, “T2D patient” also needs to be plural, “patients”

Page 2, Lines 16/17: Would recommend rephrasing the sentence so it does not start with a number. Perhaps something like this: “To examine the inner narratives about disease management and what they believe about the disease, 24 T2D patients and 10 healthcare professionals were recruited.”

Page 2, Line 25: In the abstract results section, “to disease management” should be changed to “for disease…”

Page 3, Lines 6/7: “2 folds” should be “2-fold” singular.

Page 3, Line 13: HCP should be plural (HCPs) when referring to more than one HCP.

Continued minor edits (e.g., plural vs singular, missing words, extra words) evident and need to be addressed.

Introduction:

Page 3, Lines 26-28: Would suggest not talking about publishing the procedures, but the aim of the paper is sharing the results related to…using the protocol previously published (11).

Page 3, Line 28: Grounded should not be capitalized.

Methods:

Pages 5-6, Tables 1 and 2: I am concerned about the confidentiality of data when there are cells that have less than 5 participants. These may be identifiable. Please revise, could use <5 for those cells, or report in more broad categories.

Page 7, Line 5: This paper uses a lot of abbreviations and acronyms. I would suggest not abbreviating in-depth interviews and removing the IDI abbreviation. It is not commonly used. Please attend to these revisions throughout the paper.

Data Collection and Analysis:

Reviewer 2 suggests adding information on constant comparative analysis as this is key to grounded theory. I would agree, as it currently stands, your methods could easily reflect a descriptive qualitative study.

Discussion:

The discussion section is very short and provides only a summary of the results as it currently appears. I would put the “comparison to literature” section after the discussion summary, or perhaps amalgamate these sections and then present strengths and limitations. Authors also need to discuss the impacts/relevancy for policy and practice. What is needed is the “So what?”

Strengths and Limitations:

Some of the information included here may be able to be highlighted/moved to the discussion section.

I also find that all of the results are heavily focused on medications and use of the same. Self-management is a much bigger topic than only medications. It should include eating habits/nutrition, exercise, and mental health and wellbeing, the latter which is addressed somewhat. I would encourage that if there is data, to more fully report the results. If there is no data, perhaps this is a study of self-management of medications and some of the wording and title may need to reflect that. You may also want to delineate this as a limitation in your limitations section.

References:

Reviewer 1 discusses the need for more updated references. I would agree that some of the references are fairly old, particularly those that provide statistics, or other components of self-management. We do understand that there are some seminal references, particularly around analysis and design. Please review and revise the references accordingly.

Reviewers' comments:

Reviewer's Responses to Questions

**Comments to the Author**

1. Is the manuscript technically sound, and do the data support the conclusions?

Reviewer #1: Yes

Reviewer #2: Yes

2. Has the statistical analysis been performed appropriately and rigorously? 

Reviewer #1: Yes

Reviewer #2: N/A

3. Have the authors made all data underlying the findings in their manuscript fully available?

Reviewer #1: Yes

Reviewer #2: Yes

4. Is the manuscript presented in an intelligible fashion and written in standard English?

Reviewer #1: Yes

Reviewer #2: Yes

5. Review Comments to the Author

Reviewer #1: I have gone through your document and I got it very good and sound able. I strongly suggest you the authors collect and then aggregate comments I have mentioned on the recommendation paper. Hoping you are proofreading the document so that a lot of grammatic and spelling disorganizations are modified.

Reviewer #2: This is an excellent qualitative research done by grounded theory. based on the data in their hand, the authors explained the relationship between self-management practices of Type 2 Diabetic patients and the inner psychological dispositions that affect the practice. One of the main strengths of this study is that it included variety of data sources to get full explanation of issues for all people in the given context. Participants and their recruitment process are well explained and appropriate to the study. The authors clearly described the data collection process, including recording and they collected the data using variety of qualitative techniques, which is very essential when it comes to grounded theory. The authors were responsive to most of the previous reviewers’ feedback. Overall, the methods, interpretation, and communication of the findings are relevant and reasonable.

Major comments

1, even though the authors adequately explained which purposive sampling technique did they used, they did not mention anything about which criteria/rule of saturation have they used; in order to know when to stop further data collection and start to count the number of people who had participated and eventually know the final sample size. Hence, I would like to ask the authors when was the point where they feel reasonably confident that they have saturated a particular source of information to the point of redundancy or when was the point where they feel that the incoming data have adequately answered their research questions?

2, page 7, line 4 “… Ultimately, 24 patients and 10 HCPs took part in in-depth interviews (IDI) conducted between…”

the stated data collection technique that the authors used for the 10 HCPs should not be named an ‘in-depth interview’. Because, the HCPs were not being interviewed about themselves (they are not a diabetic patients). In-depth interview is all about the participants themselves. Therefore, I would recommend it be changed to a ‘key informant interview’, as they have been interviewed only because of their position or due to the fact that they are well aware about the disease instead of being patients.

3, I wonder how the authors decided on how many focus groups to form? it is generally recommended to conduct at least two FGDs for a certain defining demographic variable. Was it conducted based on gender? Having complication or not? any other variables? Or by simply dividing the number of patients in to two randomly?

4, on the data collection, data management, and analysis section, I recommend mentioning the use of constant comparative analysis. As the research is done via grounded theory, it is basically expected to do simultaneous collection and analysis of the data.

Minor comments

1. I share a comment made by a previous reviewer which is

“There is no full form of the abbreviation “IDM” in the verbatim. The verbatim needs to be given demographic details like for example: “35-year-old T2D, self-employed male, FGD participant””.

However, I could not find this change in the post comment Manuscript Draft. I apologize if I missed this or if I am having the pre comment Manuscript Draft. Can the authors please double check to confirm they included this information?

2. page 9, line 14 this should be “…indicates that most…” rather than “…indicate that most...”

3. page 11, Line 11 Remove “the” before “… taking medication in accordance….”

4. page 11, Line 26 add “of” after “… People with diabetes recognize the importance….”

5. page 19, line 4 there is no space between two the words “….implementand appraise…”

6. page 7, line 4 This should be “….in the in-depth interviews (IDI) conducted…” rather than “…in in-depth interviews (IDI) conducted….”

Thank you,

6. PLOS authors have the option to publish the peer review history of their article (what does this mean?). If published, this will include your full peer review and any attached files.

Reviewer #1: **Yes: **Dejene Tsegaye Alem

Reviewer #2: **Yes: **Meron Asmamaw Alemayehu

---

## [Author Response · Author response to Decision Letter 0]

15 Feb 2021

We would like to thank all reviewers for examining our manuscript and providing insightful comments. We have responded to each your comments in the "Response to Reviewers" file.

---

## [Editor Report · Decision Letter 1]

23 Feb 2021

PONE-D-19-33966R1

The Internal Realities of Individuals with Type 2 Diabetes – Psychological Disposition in Self-Management Behaviour via Grounded Theory Approach

PLOS ONE

Dear Dr. Yogarabindranath Swarna Nantha,

Thank you for submitting your manuscript to PLOS ONE. After careful consideration, we feel that it has merit but does not fully meet PLOS ONE’s publication criteria as it currently stands. Therefore, we invite you to submit a revised version of the manuscript that addresses the points raised during the review process.

Thank you for submitting your revised manuscript. We have reviewed the same and require revisions as requested below. All of revisions should be addressed and submitted using track changes and line numbers in the manuscript. We look forward to receiving your revised manuscript. 

We look forward to receiving your revised manuscript.

Kind regards,

Nelly Oelke

Academic Editor

PLOS ONE

Journal Requirements:

Additional Editor Comments (if provided):

Thank you for submitting your revised manuscript. The revised manuscript has been reviewed and we would ask that you make additional revisions to the manuscript based on the following suggestions:

1. There are many lines in the manuscript that only have a word or two. The formatting needs to be adjusted.

2. There are many specific comments that still require they be addressed. They are listed as follows:

• Short title in your manuscript still does not match that in the introductory form

• Page 2, second and ninth lines, results section, and conclusion section of the abstract please spell out type 2 diabetes or consistently use the abbreviation.

• Page 2, methods in abstract should say “a” grounded theory approach instead of “the.” Also would suggest not capitalizing grounded theory.

• Page 2, second line of methods in abstract remove T2D

• Page 2, last line of results should read motivational “and” follow through

• Page 4, second line under design has an extra bracket in the middle of the line.

• Page 4, 3rd line in the design section would consider using “individuals” instead of people.

• Page 4, suggest changing “good spread” to diverse sample.

• Page 5, Line 6, “comprising” should be changed to “comprised”

• Tables 1 and 2 still have some cells with less than 5 participants (e.g., ethnicity – Chinese – 1) Please revise accordingly.

• Page 7, top of the page, numbers reported on number of participants should go before Tables and with the demographic data.

• Bottom of page 7, “remain” needs to be “remained”

• Bottom of page 7, top of page 8: the topic guide you have already talked about and referenced as reference 11. Suggest adding the interview technique above as well so this isn’t repeated.

• Text at bottom of page 7 and top of page 8 is in a different font.

• Page 8 – why were two qualitative software packages used?

• Page 8 – analysis still does not contain a lot of information on grounded theory approach (e.g., constant comparative analysis).

• Page 8 – not sure if trustworthiness can be made certain; it can certainly be enhanced and facilitated.

• Results section – you have changed much of language to people with T2D. This is a generalization and your results only apply to your participants. You will need to address and make appropriate changes.

• Page 12 – two thirds of the way down, “only during at the initial phases of their disease.” This phrase needs to be revised.

• Page 13, first paragraph, “They believe that it is necessary to consume medications lifelong…” Reword as “They believe that it is necessary for lifelong medications…” for clarity and flow.

• Page 15, second last paragraph, “appear” should be “appears.”

• Page 21, paragraph two, “we emphasized on inclusivity…”, remove “on.”

• Acknowledgements should have the “e’ in the word. Please replace the same.

3. In addition to using track changes on your the next revised version of the manuscript, please use line numbers in the document.

---

## [Author Response · Author response to Decision Letter 1]

1 Mar 2021

We have uploaded our responses in the form of a "Response to Reviewers" file

---

## [Editor Report · Decision Letter 2]

16 Mar 2021

PONE-D-19-33966R2

The Internal Realities of Individuals with Type 2 Diabetes – Psychological Disposition in Self-Management Behaviour via Grounded Theory Approach

PLOS ONE

Dear Dr. Yogarabindranath Swarna Nantha,

Thank you for submitting your manuscript to PLOS ONE. After careful consideration, we feel that it has merit but does not fully meet PLOS ONE’s publication criteria as it currently stands. Therefore, we invite you to submit a revised version of the manuscript that addresses the points raised during the review process.

Thank you for resubmitting your revised manuscript. Please address all comments below. 

We look forward to receiving your revised manuscript.

Kind regards,

Nelly Oelke

Academic Editor

PLOS ONE

Journal Requirements:

Additional Editor Comments (if provided):

Thank you for the re-submission of your manuscript and attending to many of the requirements of the reviews. There are still a few issues that need to be addressed.

1. Response and actions, if appropriate, to the comment re: retracted references.

2. Tables 1 and 2 still have cells with less than 5 participants. Please review and mark all those that are under 5 as <5.

3. There are still some areas with extra space and lines. Please remove and ensure your manuscript is consistent throughout. Also there are still a few lines that have only a few words and then go to the next line. Please remove the same.

---

## [Author Response · Author response to Decision Letter 2]

16 Mar 2021

We would like to thank the PLOS One for an excellent peer review and editorial process. We apologize for not being thorough with our previous revision process. In this current submission, we have taken every step possible to ensure that all comments have been addressed as described in the Response to Reviewer document. Please let us know if additional revisions are required.

---

## [Editor Report · Decision Letter 3]

23 Mar 2021

The Internal Realities of Individuals with Type 2 Diabetes – Psychological Disposition in Self-Management Behaviour via Grounded Theory Approach

PONE-D-19-33966R3

Dear Dr. Yogarabindranath Swarna Nantha,

We’re pleased to inform you that your manuscript has been judged scientifically suitable for publication and will be formally accepted for publication once it meets all outstanding technical requirements.

Kind regards,

Nelly Oelke

Academic Editor

PLOS ONE
---

## [Editor Report · Acceptance letter]

25 Mar 2021

PONE-D-19-33966R3 

The Internal Realities of Individuals with Type 2 Diabetes – Psychological Disposition in Self-Management Behaviour via Grounded Theory Approach 

Dear Dr. Swarna Nantha:

I'm pleased to inform you that your manuscript has been deemed suitable for publication in PLOS ONE. Congratulations! Your manuscript is now with our production department. 

Kind regards, 

on behalf of

Dr. Nelly Oelke 

Academic Editor

PLOS ONE